# Opinion: Atmospheric Multiphase Chemistry: Past, Present, and Future

Jonathan P.D. Abbatt[1], Akkihebbal R. Ravishankara[2]

[1]Department of Chemistry, University of Toronto, Toronto, ON, Canada M5S 3H6; jonathan.abbatt@utoronto.ca

[2]Departments of Chemistry and Atmospheric Science, Colorado State University, Fort Collins, CO, USA 80523; a.r.ravishankara@colostate.edu

**Abstract**

Multiphase chemistry occurs between chemicals in different atmospheric phases, typically involving gas-solid and gas-liquid interactions. The importance of atmospheric multiphase chemistry has long been recognized. Its central role extends from acid precipitation and stratospheric ozone depletion, to its impact on the oxidizing capacity of the troposphere, and to the roles that aerosol particles play in driving chemistry-climate interactions and affecting human health. This opinion article briefly introduces the subject of multiphase chemistry and tracks its development before and after the start of *Atmospheric Chemistry and Physics*. Most of the article focuses on research opportunities and challenges in the field. Central themes are that a fundamental understanding of the chemistry at the molecular level underpins the ability of atmospheric chemistry to accurately predict environmental change, and that the discipline of multiphase chemistry is strongest when tightly connected to atmospheric modeling and field observations.

**1 Introduction**
When *Atmospheric Chemistry and Physics* was launched over two decades ago, Abbatt was invited to submit an
article to the inaugural issue, which addressed the kinetics of the multiphase reaction between $SO_2$ and $H_2O_2$ on ice
surfaces (Clegg and Abbatt, 2001):
$SO_2 + H_2O_2 \rightarrow H_2SO_4$     R1
This contribution built upon concepts described in a review article published a few years earlier by Ravishankara,
which presented a conceptual view of tropospheric heterogeneous and multiphase chemistry (Ravishankara, 1997).
As part of the Special Issue entitled *20 Years of Atmospheric Chemistry and Physics*, both authors value the current
opportunity to contribute to the overall goal of the special issue "to reflect on the developments of the field of
atmospheric chemistry and physics in the last 20 years and point to exciting directions for the future" by addressing
the evolution of the field of atmospheric multiphase chemistry.  Specifically, this paper will briefly describe
multiphase chemistry, its origins, and the progress made in the past twenty years since the inauguration of
*Atmospheric Chemistry and Physics*. It then focuses in depth on future research opportunities and associated
challenges. For the sake of brevity, the citations in this paper are illustrative and not comprehensive. And so, the
reader is directed to in-depth reviews on specific aspects of multiphase chemistry (Ravishankara, 1997; Jacob, 2000;
Rudich, 2003; Usher et al., 2003; Finlayson-Pitts, 2003; Rudich et al., 2007; Kolb et al., 2010; George and Abbatt,
2010; Abbatt et al., 2012; McNeill et al., 2012; Poschl and Shiraiwa, 2015; Herrmann et al., 2015; McNeill, 2015;
Burkholder et al., 2017; Pye et al., 2020; Tilgner et al., 2021).
In the 1997 paper, Ravishankara distinguished between heterogeneous and multiphase chemistry based on the extent
of diffusion into the bulk. At that time, the term "heterogeneous chemistry" was in vogue to describe ozone hole
chemistry. Over the years, it has become clear that diffusion depths vary continuously from solid-like substrates to
dilute water solutions. Therefore, in this article, we use the term "multiphase chemistry" to refer to all chemistry that
involves more than one phase. Interfacial chemistry falls under this umbrella, with interfaces invariably present
when more than one phase is present. We note that "heterogeneous chemistry" is a useful term to describe
exclusively interfacial processes (Svehla, 1993), such as for the reactions of gas phase molecules and atoms on solid
material such as metallic or mineral catalysts. Similarly, "bulk chemistry" refers to chemistry that occurs mainly in
only one phase.  In this article, our focus is primarily on processes involving the gas phase interacting with
atmospheric condensed phases, so we do not describe in-depth advances in the associated chemistry that takes place
in the bulk phase.
One underlying theme in the paper is that understanding multiphase processes at the molecular level improves our
ability to accurately predict atmospheric change, which in turn aids in developing sustainable environmental policy
and practices.  Positive impacts arise across multiple fields, from climate and air quality to human health and
ecology.  Another theme is that multiphase chemistry studies are most impactful when closely connected to the
entire atmospheric science field, noting the interrelated nature of fundamental chemistry, field measurements, and
atmospheric modeling that together constitutes the "three-legged stool" model of our field (see Figure 1) (Abbatt et
al., 2014). Multiphase chemistry studies should be conducted to guide, interpret, and encourage field observations,
and to quantitatively inform atmospheric models.

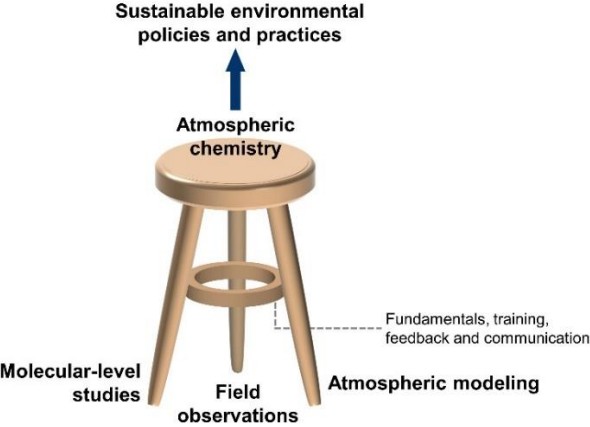


**Figure 1:** The atmospheric chemistry three-legged stool, where the legs are: i) atmospheric modeling, ii) field
observations, and iii) molecular-level studies via experimental and theoretical investigations of gas- and multiphase
chemistry.  The support ring represents the value arising from collaboration, training, and feedback that occurs
across the field, along with the need to focus on fundamental science throughout. (Figure credit: Zilin Zhou)
**2 What is atmospheric multiphase chemistry?**
Multiphase chemistry involves interactions of chemical species present in two or more atmospheric phases,
including gas-solid, gas-liquid, and liquid-solid processes.  These interactions typically require both chemical
reactions and mass transfer, i.e., the movement of a molecule within a phase or from one phase to another.  Also
important are processes in the interfacial regions, which are the thin transition zones from one phase to another.
To illustrate a multiphase process, consider R1, a key reaction leading to acid precipitation.  Gas-phase reactants
must first partition to the condensed phase, such as an aerosol particle, cloud water droplet, or ice crystal. Once
molecules collide with the interface, intermolecular forces promote adsorption for a short period, during which they
can diffuse and react via an interfacial process (see Figure 2).  If diffusion into the bulk is sufficiently fast, they can
also react in the bulk.  In the case of R1 in liquid water, dissolved $SO_2$ forms $HSO_3^-$, which can be oxidized by
dissolved $H_2O_2$ and $O_3$ to form sulfate (Hoffmann and Edwards, 1975; Penkett et al., 1979).  When the substrate is
ice instead of liquid water, the reaction proceeds either at the gas-solid ice interface or within a thin, liquid-like layer
prevalent on the ice's surface below its melting point (Girardet and Toubin, 2001; Abbatt, 2003).
The interface is involved in all multiphase processes, participating in mass transfer and reactivity. For a low-
volatility material, such as a bare mineral or metal, there is a step function drop in the density moving from the
condensed phase to the gas phase. A solid's interface, roughly one molecule (or atom) deep, can promote
heterogeneous chemistry that occurs in a two-dimensional space (Tabazadeh and Turco, 1993). However, solid
particles in the atmosphere, such as soot or mineral dust, frequently have mobile, semi-volatile molecules, such as
water, sulfuric acid, and organics, deposited as multilayer-thick films or islands. In these cases, the multiphase
chemistry is likely occurring in a three-dimensional space involving chemical interactions with not only the solid
substrate but also the liquid coatings (Worsnop et al., 2002; Poschl et al., 2007). For cloud droplets and liquid
aerosol particles, which are very dynamic with large fluxes of substrate molecules being rapidly exchanged between
the phases, the interfacial region is also thicker than it is for a pure solid (Gopalakrishnan et al., 2005). When the
underlying substrate is a liquid, the depth to which a molecule diffuses from the interface into the bulk before
reacting is called the reacto-diffusive length, which can be very short for species reacting close to the interface and
extending to the size of an aerosol particle for reactions that are aerosol-volume limited (see Figure 2) (Hanson et
al., 1994). It is not easy to determine whether a reaction is occurring exclusively in a two-dimensional manner at an
interface, where the concept of reacto-diffusive depth breaks down because of different energetics and solvation
than in the bulk phase. For particle chemistry, the reaction kinetics dependence on particle size and reactant
concentration can provide some information. Nevertheless, even in reactions which exhibit Langmuir-Hinshelwood
kinetics (Poschl et al., 2001), the depth of the reactive region at the surface of a particle is not clear for liquids or
coated solids.



**Figure 2:** Multiphase reactions occur at the interface and within the bulk of condensed phase materials. Bare solids undergo heterogeneous chemical reactions at the interface. High-viscosity liquids may have sufficiently short reacto-diffusive lengths to restrict reactivity to the surface of the particle. As the viscosity lessens and the reacto-diffusive length deepens, multiphase chemistry can occur deeper into the bulk phase. For reactions that proceed in the bulk, some component of the reaction may also simultaneously occur at the interface (as indicated by the dashed reaction arrow). As noted in the text, the concept of the reacto-diffusive length, which is formally calculated from the bulk diffusion and rate constants, may break down in the interface region. (Figure credit: Zilin Zhou)

**3 How does multiphase chemistry differ from gas-phase chemistry?**

Although thermodynamically allowed, reactions between neutral closed-shell molecules are slow in the gas phase because of large reaction barriers. However, the rates of multiphase reactions involving the same reactants (or suitably altered versions in the condensed phase) can be larger than in the gas phase, either because the free energy barrier to reaction is lowered or because the concentrations of reactants are enhanced in the condensed phase. To illustrate, $SO_2$ and $H_2O_2$ do not react efficiently in the gas phase, but oxidation with dissolved $H_2O_2$ can proceed once $SO_2$ dissolves in water and forms $HSO_3^-$. Moreover, the gas phase $H_2O_2$ is efficiently scavenged in clouds, enhancing its concentration for reaction.

Another example is the set of reactions that drive polar stratospheric ozone depletion (Solomon, 1999). Chlorine reservoir compounds such as $ClONO_2$ and HCl do not react rapidly in the gas phase. However, HCl partitions strongly to polar stratospheric clouds by either adsorbing to their surfaces or dissolving within them. For example, it is likely that both adsorbed and dissolved HCl dissociates on/in ice to form chloride ions, which are reactive with $ClONO_2$:

$$ClONO_2 + HCl \rightarrow Cl_2 + HNO_3 \qquad R2$$

leading to the formation of $Cl_2$, which is then released to the gas phase and is readily photolyzed, forming radicals that catalyze gas phase ozone destruction in the Antarctic ozone hole. Also, $ClONO_2$ can be protonated in strongly acidic cloud droplets, or it may dissociate to form $Cl^+$ and $NO_3^-$ (although there is no direct experimental evidence yet for forming $Cl^+$ as an independently solvated species). Other possible mechanisms include a concerted reaction between $Cl^-$ and $ClONO_2$ to produce $Cl_2$ (Bianco and Hynes, 1999). It is unclear if such a reaction is truly an SN2-type process or if it can occur in a cage in the condensed phase.

Another distinguishing feature of multiphase chemistry is that it can lead to the formation of products that do not arise in the gas phase. Consider the acid-catalyzed nucleophilic reactions between sulfate and isoprene-derived epoxydiols that form organo-sulfate molecules and secondary organic aerosol (SOA) (Riva et al., 2019). Water molecules lower the transition state energies of such reactions. The solvent shell, which confines reactant partners via the so-called cage effect, can also promote novel products. For example, the formation of biologically-active secondary ozonides is facile in the condensed-phase ozonolysis of unsaturated fatty esters and triglycerides, arising from reactions of Criegee and carbonyl intermediates that form in the same solvent shell after dissociation of a primary ozonide (Zhou et al., 2019b, 2022). In the gas phase, the solvent shell is essentially absent (except for some

specific cases, such as cluster formation with $H_2O$), so the carbonyl and Criegee intermediates fly apart, and
secondary ozonides do not form so readily.
Lastly, some multiphase reactants, such as transition metal ions, are absent in the gas phase. More generally, ion-ion
and ion-molecule regions play a much greater role in condensed phase chemistry than in tropospheric and
stratospheric gas phase chemistry, leading to a wide variety of novel reaction pathways, with R1 being an excellent
example.

**4 Early studies of atmospheric multiphase chemistry**

Studies of aerosol and cloud chemistry have proceeded in concert with the development of the wider atmospheric
chemistry field with many of the concepts of coupled reactivity and mass transfer initially developed by the process-
oriented chemical engineering community (Dankwerts, 1970).  Interest in multiphase reactions arose via the
profound ways these sparse aerosol particles and cloud droplets can alter gas phase composition. Also, as described
in more detail below, it is now evident that the reverse is important, i.e., the gas phase alters the condensed media
with important environmental consequences.  Many aerosol and multiphase reaction studies were initially performed
to develop parameterizations for atmospheric modeling. Even though this is still a major goal, much more effort is
now given to understand the physico-chemical processes, which is essential for predictive capabilities.
The idea of reactive chemistry in hydrometeors goes back to the late 1960s and 1970s and possibly earlier. As in
much of science, it is hard to pinpoint a specific paper that expounded this idea. The expectation that the $SO_2$
pollutant can be oxidized to sulfuric acid in the water droplets in the atmosphere spurred many studies, hypotheses,
and definitive results. Early studies examined the potential oxidation of $SO_2$ in the liquid phase via a variety of
oxidants, including $O_3$, $H_2O_2$, and $NO_2$ (Hoffmann and Edwards, 1975; Erickson et al., 1977; Schroeder and Urone,
1978; Penkett et al., 1979). As well, modeling studies showed the feasibility of such oxidation reactions occurring in
the atmosphere (Jacob and Hoffmann, 1983; and references therein). The Great Dun Fell experiment observationally
established that $SO_2$ is indeed oxidized in the troposphere via reactions in cloud droplets (Choularton et al., 1997).
Similar multiphase reactions in the stratosphere were sometimes evoked but never pursued with great vigor until the
spectacular occurrence and subsequent explanation of the ozone hole (Solomon, 1999) and a multitude of laboratory
studies showing that indeed there can be chlorine activation (Molina et al., 1987; Tolbert et al., 1988; Leu, 1988;
Hanson and Ravishankara, 1992; and many others).
Additional pioneering atmospheric multiphase chemistry studies arose from aerosol composition measurements
conducted over a half-century ago.  Specifically, continental aerosol particles always contain a measurable quantity
of ammonium, indicating the uptake of gas-phase ammonia to acidic particles (Lee and Patterson, 1969; Kadowaki,
1976).  Furthermore, particulate chloride levels in the marine aerosol are depleted relative to their seawater
abundance, replaced by sulfate or nitrate (Junge, 1956; Martens et al., 1973). This process was long thought to be
the major source of gas-phase chlorine, whereby gaseous HCl is displaced from NaCl particles via the uptake of gas-
phase strong acids :
$H_2SO_4 + 2\ NaCl \rightarrow 2\ HCl + Na_2SO_4$    R3
Another early example of halogen chemistry demonstrated that volatile iodine species are released when dissolved
oceanic iodide is exposed to either ultraviolet light or ozone (Miyake and Tsunogai, 1963; Garland et al., 1980).
This multiphase chemistry is important for the dry deposition of ozone and the release of iodine into the atmosphere
(Carpenter et al., 2013):
$O_3 + H^+ + I^- \rightarrow HOI + O_2$    R4
$HOI + I^- + H^+ \rightarrow I_2 + H_2O$    R5
After these early studies, many additional important tropospheric multiphase chemical processes were identified
prior to the launch of *Atmospheric Chemistry and Physics*. Some examples, which all address gas-particle and cloud
droplet interactions, include: i) the formation of reactive halogen species in the boundary layer (Finlayson-Pitts,
2003; Simpson et al., 2015), ii) the uptake of tropospheric gases by mineral dust, especially nitric acid (Hanisch and
Crowley, 2001; Usher et al., 2003), iii) the scavenging of trace gases, such as nitric acid and small oxygenated
VOCs, by snow and ice crystals in the free and upper troposphere (Abbatt, 2003), iv) the impact of aqueous cloud
and aerosol chemistry on gas phase $HO_x$ levels (Chameides and Davis, 1982; Calvert et al., 1985; Jacob, 1986;
Lelieveld and Crutzen, 1991), v) conversion of $N_2O_5$ to $HNO_3$ on tropospheric aerosol, with impacts on the $NO_x$
budget (Dentener and Crutzen, 1993), vi) uptake of $HO_2$ to aerosol (Mozurkewich et al., 1987; Martin et al., 2003),
and vii) multiphase conversion of $NO_2$ to HONO (Finlayson-Pitts et al., 2003).  A critical point is that each of these
multiphase processes affects the oxidizing capacity of the troposphere, frequently through modification of radical
budgets and occurring via gas-aerosol or gas-droplet interactions.  For example, these processes initiate oxidation in
urban atmospheres through HONO photolysis, drive Arctic boundary layer ozone and mercury depletion via gas-
phase halogen chemistry, and modulate the global oxidizing capacity via $N_2O_5$ or $HO_2$ loss on aerosol particles.
Additional work in the multiphase world at this time involved a wide variety of condensed-phase photochemistry
studies, for example involving the interactions of light with nitrate (Zepp et al., 1987), which can lead to the release
of $NO_x$ to the gas phase, and with transition metal ion complexes (Faust and Zepp, 1993).
**5 Progress in the past twenty years**
Two major developments profoundly influenced multiphase chemistry. First was the recognition of the importance
of aerosol particles in changing the radiative balance of the Earth system, with impacts on climate. The second was
the continued recognition of the deleterious effects of particulate matter on human, animal, and ecosystem health.
These two fields, climate change and air quality, have provided the impetus (and resources) for the development of
the field.  As a result, additional research emphasis in the multiphase chemistry community was given at the turn of
the 21$^{st}$ century to assess the impacts that arise on the composition of the particles.
Once inhaled, particles harm human health (Landrigan et al., 2018; Murray et al., 2020), with recent studies
implicating the secondary component of the particles in negative health outcomes (Pye et al., 2021).  Research in the
past two decades has focused strongly on the formation of SOA (Kroll and Seinfeld, 2008; Hallquist et al., 2009;
Ziemann and Atkinson, 2012; Shrivastava et al., 2017b). SOA formation has required better knowledge of the
kinetics and mechanisms of gas phase oxidation of SOA precursors (Crounse et al., 2013; Ehn et al., 2014). It has
also needed a more complete understanding of gas-particle nucleation processes (Kulmala et al., 2014; Trostl et al.,
2016; Xiao et al., 2021), volatility (Pankow, 1994; Donahue et al., 2011), and condensed-phase reactions that occur
within aerosol particles. Specifically, volatility and multiphase reactivity can be coupled, as illustrated by the
formation of high molecular weight, low volatility species within particles (Kalberer et al., 2004).  While such
oligomers and highly oxygenated species may also form in the gas phase (Bianchi et al., 2019), they arise via a
variety of reactions involving pairs of organic reactants, frequently forming esters and acetals/hemiacetals in the
condensed phase (Tobias and Ziemann, 2000; Surratt et al., 2006; DeVault and Ziemann, 2021). These reactions
may be acid-catalyzed (Jang et al., 2002). Also, multiphase oxidation by gas-phase oxidants can increase the average
oxidation state of organic aerosol particles (Kroll et al., 2011) via a series of reactions that initially functionalize and
eventually fragment the component molecules (Moise and Rudich, 2000; Molina et al., 2004; George et al., 2007;
Kroll et al., 2009).  Oxidation leads to a more soluble particle that increases its rate of wet deposition.  In addition to
forming organic aerosol via gas-to-particle conversion, they are produced from the evaporation of cloud droplets.
Oxidation processes occur within cloud droplets (Herrmann et al., 2015), producing more oxidized organics that
yield oxygenated aerosol upon evaporation. Similar reactions, proceeding at much higher organic reactant
concentrations, can also occur within the aqueous component of tropospheric aerosol (Blando and Turpin, 2000).
Tightly connected to SOA formation and modification processes are the condensed phase viscosity and phase state,
which set mixing times within particles and are dependent on relative humidity and temperature (Virtanen et al.,
2010; Koop et al., 2011; Renbaum-Wolff et al., 2013). Organic particles are likely glasses in the cold free
troposphere (Shiraiwa et al., 2017b), which may affect SOA formation and growth, and restricts the degree to which
heterogeneous oxidation can affect the aerosol composition. The particles are liquids in warm, wet boundary layers,
with the full particle volume involved in partitioning with gas phase molecules.  The large variation in molecular
diffusion coefficients and associated mobility determines where chemical reactions are important in the particles,
from two-dimensional processes that occur solely at the gas-particle interface to three-dimensional chemistry with
reactivity at the interface and deeper in the bulk (see Figure 2). Overall, diffusion is a key parameter for determining
whether a reaction is surface-area-limited or volume-limited (Hanson et al., 1994).
Multiphase chemistry also leads to the formation of secondary inorganic aerosol. For example, the hydrolysis of
$N_2O_5$ converts $NO_x$ to $HNO_3$; the gas-particle partitioning of $HNO_3$ is then controlled by temperature, relative
humidity, and ammonia levels.  Also, particulate sulfate is rapidly formed in polluted environments through
multiphase aqueous chemistry, acting as the major formation mechanism in cloud-free settings (Cheng et al., 2016;
Wang et al., 2016). Potential routes for fast sulfate formation in deliquesced particles include: the role of ionic
strength in accelerating the rates of specific processes (Liu et al., 2020), fast interfacial chemistry (Liu and Abbatt,
2021), formation of condensed-phase oxidants through the photolysis of particulate nitrate (Zheng et al., 2020; see
also Section 6.5), and the role of specific particle-phase reactants, such as organic hydroperoxides (Wang et al.,
2019), hydroxymethanesulfonate (Song et al., 2019), and dissolved transition metal ions (Li et al., 2020b).  An
accurate quantitative assessment of these and other reaction pathways is still developing but far from complete (Liu
et al., 2021b).
As noted earlier, the need to better understand aerosol-climate interactions has also motivated multiphase chemistry
research in the past twenty years. Atmospheric processing leads to the formation of water-soluble condensed-phase
products, such as sulfate or highly oxygenated organic molecules (Jimenez et al., 2009), enhancing the abilities of
tropospheric aerosol particles to act as cloud condensation nuclei (CCN) and affecting their ability to scatter light
(Cappa et al., 2011; Moise et al., 2015).  As well, the optical properties of the fraction of organic aerosol that
absorbs near ultraviolet and visible light (i.e., atmospheric 'brown carbon' particles) are subject to change via
multiphase oxidation and condensed phase photochemistry (Laskin et al., 2015; Li et al., 2020a; Hems et al., 2021;
Schnitzler et al., 2022), potentially involving transition metals (Al-Abadleh and Nizkorodov, 2021). Although the
rates of optical property changes remain uncertain, primary brown carbon particles, as formed in wildfires, tend to
become less absorbing in the near UV and visible parts of the spectrum on the timescale of days via a variety of
multiphase aging mechanisms (Laskin et al., 2015; Hems et al., 2021), i.e., they are 'bleached.' The diminution of
light absorption is in accord with field observations (Forrister et al., 2015).
Multiphase chemistry can also affect the properties of ice nucleating particles (INPs) by both gas-solid and liquid-
solid interactions, noting that INPs can act in the deposition mode where water vapor forms ice directly on solid
substrates and in the immersion mode where a solid particle immersed in supercooled water leads to nucleation
(Kanji et al., 2017). For example, mineral dust can have its IN activity decreased by condensation of involatile
materials, such as sulfate or by cloud processing (Sullivan et al., 2010b; Kilchhofer et al., 2021), and strong acids
can react with carbonate-containing minerals, leading to particles that are less IN-active in the deposition mode but
more active in the immersion mode (Sullivan et al., 2010a).  Such effects can also arise when different gas and
liquid species are exposed to volcanic ashes (Maters et al., 2020; Fahy et al., 2022). Oxidation reactions can also
occur so that efficient biological INPs, such as pollen fragments, lose activity upon oxidation by OH radicals,
probably by morphological changes of surface proteins and carbohydrates (Gute and Abbatt, 2018). The
mechanisms involving all these interactions are very complex. In the case of mineral dusts, immersion INP activity
can be changed by surface modification, ion exchange, adsorption of solutes such as ammonium, and acid
dissolution (Sihvonen et al., 2014; Kumar et al., 2019; Yun et al., 2021).
**6 The future of atmospheric multiphase chemistry studies**
**6.1 Multiphase chemistry at the interfaces of the atmosphere**
There are exciting opportunities for applying the conceptual, instrumental, and modeling tools developed for
multiphase chemistry to understand chemistry occurring at the interfaces of the atmosphere with other
environmental domains.
Consider the interface of the atmosphere and the ocean, where the sea-surface microlayer (SML) is a thin layer of
ocean water that has enhanced concentrations of biological detritus, surface-active compounds, and gel-like
substances (Cunliffe et al., 2013). Recognizing that individual molecule surrogates of the SML only capture specific
aspects of the chemistry, experimental designs now involve either genuine seawater or water samples with
significant biological components (Prather et al., 2013; Schneider et al., 2019). While we know that the SML can
affect the composition of primary marine aerosol, an open question is the degree of chemical processing by
photosensitization in the SML or by gas-surface heterogeneous oxidation, yielding volatile species that contribute to
marine SOA formation (Donaldson and George, 2012; Rossignol et al., 2016; Mungall et al., 2017; Croft et al.,

286    2019).

Another key role of multiphase reactions is in dry deposition processes on the ocean (e.g., see R4 and R5),
vegetation, the built environment, and land surfaces (Garland et al., 1980; Fowler et al., 2009; Kavassalis and
Murphy, 2017; Tuite et al., 2021). Deposition is a critical step that controls removing chemicals from the
atmosphere. Yet, this process is a parameterization in models. Deposition in many environments needs to be
predictive, which demands molecular-level understanding and quantification. This process is essentially a
multiphase process that should be broken down into physico-chemical steps, which can be independently measured
and understood.
Indoor environments, with their vast surface area-to-volume ratios, are another example of poorly explored
multiphase processes (Morrison, 2008; Abbatt and Wang, 2020; Ault et al., 2020). Contrary to the outdoor
environment, where aerosol particles may remain suspended for days to weeks, the indoor air-exchange timescale is
on the order of an hour or two. While this lessens the potential for gas-particle chemistry, multiphase chemistry
occurs over much longer timescales on fixed indoor surfaces. For example, $O_3$ is efficiently lost via dry deposition
so that its mixing ratios are considerably lower indoors than outdoors (Weschler, 2000). This produces VOCs
(Wisthaler and Weschler, 2009) and modifies the composition of sorbed molecules, in some cases forming species
more toxic than their precursors (Pitts et al., 1978, 1980; Zhou et al., 2017). It can also lead to the formation of gas-
phase OH radicals (Zannoni et al., 2022). This source of OH from alkene ozonolysis is in addition to the generation
of OH from photolysis of HONO (Gomez Alvarez et al., 2013), which is partly formed by multiphase chemistry on
indoor surfaces. Indoor surfaces are a chemically complex, poorly understood environment, with input from building
materials, commercial products, humans, and cooking and cleaning activities. This chemistry is important because
humans obtain most of their chemical exposure indoors, not only via inhalation but also through direct dermal
uptake and by ingesting dust and contaminated foodstuffs (Li et al., 2019b). Lastly, the light environment indoors
can be substantially different than outside, bringing a new twist to multiphase photochemistry (Young et al., 2019).
**6.2 Multiphase chemistry and human health**
Epidemiological studies have conclusively shown that aerosol particle inhalation harms human health (Pope et al.,
2009; Landrigan et al., 2018). For example, it is well-recognized that inflammation occurs upon particle exposure
(Brook et al., 2003). The current epidemiology (empirical evidence) does not readily distinguish the specific
molecules in the particles and their formation pathways that lead to negative health outcomes, nor the toxicity
mechanism at the molecular level. Studies are currently examining oxidative stress, e.g., the reactive oxygen species
(ROS) and reactive nitrogen species (RNS), as a mechanism for negative impacts (Shiraiwa et al., 2017a). Although
there is debate over whether oxidants are largely endogenous or exogenous (Fang et al., 2022), one hypothesis is that
the biochemical balance between oxidants and antioxidants is upset by inhaling harmful species (Miller, 2020).  To
contribute to this debate, the multiphase chemistry community needs to better describe the chemistry that occurs at
the lung-air interface and the composition of respirable aerosol particles, especially the biologically active
components that contain reactive functional groups (e.g., epoxides, hydroperoxides), redox-active materials (e.g.,
quinones), and reactive oxygen species (e.g., peroxides, $HO_2/O_2^-$).  Many of these species are formed by multiphase
oxidation processes.

An associated issue is how ultrafine particles influence health. These particles have been shown to be taken directly
to the bloodstream and even move to the brain (Oberdorster et al., 2004; Maher et al., 2016). Though the chemistry
involved is not the multiphase chemistry discussed here, the interactions of the particle in the liquid phase (i.e.,
impacting biological systems) are likely important. Many of the lessons learned from studies of multiphase
processes are likely applicable to understanding such issues.
Largely unexplored until the recent COVID-19 pandemic is the impact of the atmosphere on airborne and surface-
deposited biological pathogens, including bacteria and viruses. Early work in this area included the multiphase
chemistry between $NO_2$ and proteinaceous material, motivated by its potential to drive an allergenic response
(Franze et al., 2005; Shiraiwa et al., 2012).  Gas phase $O_3$ has also been examined for its ability to affect the viability
of bacteriophages, i.e., microorganisms with a lipid envelope and RNA core similar to the structure of SARS-CoV-
2, deposited on surfaces (Tseng and Li, 2008). With the pandemic, research has accelerated into the impact of
hygroscopic growth and water content on viral viability within respiratory particle surrogates that consist of viruses
embedded in saline droplets containing surfactants, proteins, and carbohydrates. It is important to understand the
changes in the acidity of these particles, the mass transfer within them, and the precipitation of salts as the particle
water content changes (Lin et al., 2020; Oswin et al., 2022; Huynh et al., 2022).
The recent pandemic led to an emphasis on cleaning surfaces to reduce the potential for infection by fomites, i.e., via
contact with contaminated surfaces.  While cleaning agents such as chlorine bleach have well-established anti-
microbial activity, their multiphase chemistry can release gases and particles that are deleterious to human health
(Wong et al., 2017; Mattila et al., 2020). Understanding the multiphase chemistry associated with these cleaning
activities and the outcomes of using air cleaners (Collins and Farmer, 2021), is essential for establishing healthy
indoor environments.
Lastly, the pandemic prompted a re-examination of an overlooked aspect of our atmosphere that it has an as-yet-
unidentified germicidal component referred to as the Open Air Factor (Cox et al., 2021).  In particular, it was shown
many decades ago that fresh air led to better outcomes for tuberculosis patients and injured World War I soldiers
than indoor air. Historically, sending sick people to pristine environments (e.g., the seaside) was a common medical
recommendation. It is crucial to determine whether these effects are related in some way to multiphase chemistry
occurring between reactive species in the gas phase interacting with biological molecules at the surface of the
wounds and lungs.
Each of the above topics provides exciting opportunities for atmospheric chemists to collaborate with the
environmental health, medical, and toxicological communities.
**6.3 Chemistry of the free troposphere and lower stratosphere**
Although the upper troposphere – lower stratosphere region was the focus of much attention in the 1980s and 1990s
to understand the changes in ozone levels in these regions, most multiphase chemistry studies are currently
conducted at room temperature.  There is considerable motivation to re-explore chemistry at colder temperatures,
given past work that illustrated the atmospheric impacts of a strongly non-linear dependence of multiphase reactions
rates on temperature (Murphy and Ravishankara, 1994) and extensive new observations from the Atom campaigns
(Thompson, 2022) that sampled from the boundary layer to the upper troposphere over many latitudes and seasons.
As well, there is emerging evidence for organic aerosol in the lower stratosphere, likely arising from wildfire
injection, with potential influence on stratospheric ozone (Solomon et al., 2022; Strahan et al., 2022).
Organic aerosol viscosity and phase state depend on the environmental conditions (Koop et al., 2011), with semi-
solid and glassy organic particles predicted throughout much of the free troposphere (Shiraiwa et al., 2017b).  Aside
from those at the gas-particle interface, molecules in highly viscous organic particles are protected from
heterogeneous oxidation (Shiraiwa et al., 2011; Zhou et al., 2012; Shrivastava et al., 2017a). Such protection
increases the lifetimes of pollutants, e.g., brown carbon chromophores (Schnitzler et al., 2022), and lengthens
particles' oxidation timescale and wet deposition lifetime.
In addition to continuing to address the fundamentals of cloud chemistry oxidation processes, the associated
chemistry of transition metals, and the production of oxidants within cloud water and via uptake from the gas phase
(Herrmann et al., 2015), there is a particular need to also study such processes at cold temperatures, including under
supercooled water conditions. When supercooled water is frozen, solutes are excluded from the ice crystals and
become highly concentrated at grain boundaries and in liquid and liquid-like layers at the surface, potentially leading
to enhanced rates of aqueous phase chemistry. As well, the Reynolds-Workman potential (Workman and Reynolds,
1950), arising at the ice-water interface, can drive chemistry.
A key factor affected by temperature is the solubility of various atmospheric constituents. Simple Henry's law
constants and further equilibration steps that determine the overall solubilities are poorly known, especially below
room temperature.  Most of the data on the solubilities in organics goes back to chemical engineering literature that
is more than half a century old. Also, since solubilities vary according to Henry's law equilibria that vary
exponentially with temperature, the accurate temperature dependence of solubilities is essential. Acid dissociation
constants in organic acids and organic substrates are poorly known, and they determine the overall solubility of a
chemical.
**6.4 Reactive transformations of organic chemical contaminants**
Over forty years ago, it was recognized that multiphase oxidation of chemical contaminants leads to the rapid loss of
surface-bound PAHs and the formation of more toxic and potentially carcinogenic products such as nitro-PAHs and
oxygenated PAHs (Pitts et al., 1978, 1980).  These reactions occur on a variety of surfaces with light, ozone, and
$NO_2$ reactants, some via Langmuir-Hinshelwood mechanisms (Poschl et al., 2001; Mmereki and Donaldson, 2003;
Kwamena et al., 2004). Buried PAHs are protected from heterogeneous loss by a crust of unreactive products that
accumulates upon them and, when present, within viscous organic aerosol (Zhou et al., 2013, 2019a), enabling the
potential for long-range atmospheric transport (Mu et al., 2018). The chemistry of other organic contaminants,
including smoking products such as nicotine (Destaillats et al., 2006) and tetrahydrocannabinol (Yeh et al., 2022), a
few pesticides (Segal-Rosenheimer and Dubowski, 2007; Finlayson-Pitts et al., 2022), and organophosphate esters
(Liu et al., 2021a), has also been recently explored.
However, these are largely exceptions, and the multiphase fate of most chemical contaminants, especially thousands
of commercial products, has not been examined.  Indeed, the atmospheric chemistry and chemical contaminant
communities have traditionally not strongly interacted.  Although assessment of the gas phase OH reactivity is
customarily performed in environmental fate analyses (Li et al., 2019b), many commercial products have
sufficiently low volatility such that they reside primarily on surfaces or within particles.  It is important to establish
whether organic contaminants traditionally viewed as persistent are indeed unreactive with respect to multiphase
transformation.
**6.5 Understanding the role of light**
Many condensed-phase photochemical reactions proceed via indirect mechanisms where a photosensitizing
molecule absorbs light, forming reactive species such as $HO_2/O_2^-$ or $^1O_2$ (George et al., 2015). Such chemistry, first
identified for natural waters (Canonica et al., 1995), has been implicated in the daytime formation of HONO
(George et al., 2005), the photoreactions of brown carbon aerosol (Laskin et al., 2015; Hems et al., 2021), the
formation of active halogens (Reeser et al., 2009), and reactivity of black carbon (Monge et al., 2010; Li et al.,
2019c). This chemistry has been illustrated using efficient photosensitizing agents, but quantitative assessments of
atmospheric importance remain uncertain largely because the character and quantity of atmospheric photosensitizers
are not well established. Developing a tighter quantitative connection to the atmosphere will require using more
representative photosensitizers, as now being done using marine aerosol components (Ciuraru et al., 2015; Garcia et
al., 2021). The wavelengths of interest for the troposphere are in the near UV and visible part of the solar flux.
In addition to indirect sensitization, light can also lead to direct photochemistry. An important finding was that
photolysis on ice and snow surfaces was demonstrated to form $NO_x$ in midlatitudes and polar regions (Honrath et al.,
1999; Wolff et al., 2002; Domine and Shepson, 2002). This process, which likely proceeds in a wide range of
environments, is now referred to as "re-noxification" as it releases $NO_x$ from $HNO_3$ that has deposited from the
atmosphere. Other condensed-phase chemical processes of importance include the formation of oxidants from
nitrate and nitrite photolysis (Zepp et al., 1987), photolysis of condensed-phase organic hydroperoxides and other
highly oxygenated organics, and the photochemical activity of many transition metal ion complexes (Faust and
Zepp, 1993; Weller et al., 2013).  It is important to recognize that the absorption spectra and product quantum yields
of dissolved species can be different than those in the gas phase (George et al., 2015), with aqueous nitrate a prime
example (Benedict et al., 2017). The variable viscosity of organic aerosol matrices can affect photolysis rates,
products, and their temperature dependence (Lignell et al., 2014).

**6.6 Developments in Field Observational Capabilities**

Our ability to characterize atmospheric composition continues to push the field of atmospheric chemistry forward.
For multiphase chemistry, advances in analytical mass spectrometry have been transformative. Within the last
twenty years, online characterization of aerosol composition has become commonplace (Canagaratna et al., 2007),
studies of single particle composition allow us to observe the variability in mixing state and chemical diversity
(Zelenyuk and Imre, 2005; Murphy et al., 2006; Prather et al., 2008), and offline filter sampling has progressed from
the characterization of a few targeted species to non-targeted analyses using a range of mass spectral ionization
methods (Papazian et al., 2022; Ditto et al., 2022). Identifying specific molecular "markers" for organics and
functional groups is still somewhat uncertain; developing such identification would be very helpful.
The continued development of analytical techniques will enable increasingly sophisticated characterization of
aerosol particles and environmental surfaces, with the opportunity to deploy the same tools in both lab and field
settings.  However, challenges are arising as well.  Despite the rapid development of low-cost sensors, affordable
instrumentation for the long-term characterization of aerosol composition in many locations is still lacking. The
increasing sophistication of analytical instrumentation also continues to unveil the high degree of chemical
complexity present.  Whereas high-resolution mass spectrometry yields chemical formulae in real-time, there is
often the need to identify chemical structures. This suggests that we should increasingly deploy separation
techniques (e.g., chromatography, ion mobility) as front ends to our increasingly sophisticated mass spectrometric
techniques (Krechmer et al., 2016; Claflin et al., 2021).  There is also value to the expanded use of other classical
chemical speciation methods, such as infrared (Russell, 2003) and NMR (Decesari et al., 2007) characterization of
aerosol composition collected by filters. While these techniques have low time resolution, they provide
complementary quantitative and functional group information and can be inexpensively deployed for long-term
analyses in a wide range of environments. Such analyses will also help with the source apportionment of the
aerosols.
Aerosol characteristics related to multiphase chemistry can be studied with increasingly sophisticated remote
sensing techniques. These approaches have been applied for many years to polar stratospheric clouds, whose
composition and phase (via the degree of depolarization of a lidar probe) have been studied (Tritscher et al., 2021).
Another example comes from satellite measurements of solid ammonium nitrate particles in the upper troposphere,
driven by the Asian monsoon that uplifts ammonia-rich continental air (Hopfner et al., 2019).  It is important to
determine the role of these particles in ice nucleation and multiphase chemistry.

**6.7 Developments in Laboratory and Molecular Modeling Techniques**

In addition to our ability to conduct field observations, a revolution has occurred in the laboratory's analytical
methods. This is most widely apparent in applying sophisticated mass spectrometric techniques, increasingly
involving high mass resolution and a range of ionization schemes (Laskin et al., 2013). When coupled with other
analytical methods, we can now determine the physico-chemical properties of individual molecules and their
mixtures in extreme detail. For example, this approach has been taken to characterize the optical properties of
brown carbon aerosol materials (Fleming et al., 2020), the viscosity of organic aerosol (DeRieux et al., 2018), and
the structural isomers of complex organic mixtures (Krechmer et al., 2016). There are significant opportunities for
additional adoption of techniques from neighboring disciplines. As well, the use of a number of these analytical
techniques in both the laboratory and the field will enhance our ability to connect the lab to the field.
Molecular-level chemical models increasingly provide valuable insights into complex multiphase processes. For
example, important insights into the nature of the chemistry occurring on polar stratospheric cloud materials were
obtained from molecular dynamics modeling (Wang and Clary, 1996; Bianco and Hynes, 2006) and more recent
studies have addressed gas-surface interactions and the roles of solvent molecules in small molecular clusters
(Gerber et al., 2015; Fang et al., 2019; Yang et al., 2019). Whereas past computational methods only included a few
solvent molecules, current dynamics models using state-of-the-art force fields can realistically simulate partitioning,
surface adsorption constants, diffusion constants, and vapor pressures, representing an important point of contact to
the physical chemistry and chemical physics communities (Tobias et al., 2013). We also note that machine-learning
techniques are very recently being applied to molecular dynamics simulations, for example, to describe the
interactions of $N_2O_5$ with liquid water (Galib and Limmer, 2021) and the dissociation of strong acids at aqueous
interfaces (de la Puente et al., 2022). For establishing fundamental parameters that are experimentally challenging to
measure, such as the likelihood that a collision of a molecule with a particle leads to uptake by the condensed phase
(i.e., a mass accommodation coefficient), theoretical methods may be preferable to experiment in some situations.
**6.8 Grappling with chemical complexity**
Atmospheric aerosol particles and surfaces are morphologically and compositionally complex. This complexity can
be enticing from a fundamental chemistry perspective as we disentangle mass transfer, phase separation, and
reactivity. However, it can impede the development of an accurate, quantitative description required to inform an
atmospheric model. It can also be constraining if we study the detailed chemistry and lose sight of its overall impact
on climate, air quality, or ecosystem health.
With enough care, the rate constant for a gas-phase, radical-molecule reaction can be measured with 10-20%
accuracy (Cox, 2012). Atmospheric modelers rely upon this confidence level as they assess their predictions. It is
humbling to consider the accuracy of the available multiphase kinetics data for the modeling community. Take for
example the reaction of $N_2O_5$ with tropospheric aerosol, which has been long known to impact $NO_x/NO_y$ and active
chlorine levels, with a secondary influence on OH, $O_3$, and $CH_4$ (Dentener and Crutzen, 1993). Although studies
started in the 1980s, new mechanistic insights on $N_2O_5$ heterogeneous reactivity are still arising (Sobyra et al., 2019;
Karimova et al., 2020). Laboratory reactive uptake coefficients for the hydrolysis of $N_2O_5$ vary over one-to-two
orders of magnitude, with larger values reported for aqueous particles composed of sulfate or soluble organics, and
lower values for particles with less soluble organics and nitrate (Burkholder et al., 2020). Likewise, uptake
coefficients inferred from field measurements or with genuine ambient particles vary by roughly an order of
magnitude compared to those measured with laboratory surrogates (Brown et al., 2006; Bertram et al., 2009; Abbatt
et al., 2012; Phillips et al., 2016; Tham et al., 2018). The discrepancies between field and lab studies are
undoubtedly due to complex and variable particle composition and phase state. Simply put, unlike the case with gas
phase reactions, one of the "reactants" in this gas-particle reaction is highly variable. This complexity is exacerbated
by the changes in the composition (including acidity), mixing state, and water content of the particle as it resides in
the atmosphere. Added complexity arises from the differences in composition that occur in the bulk of particles and
droplets compared to their interfacial composition (Wingen and Finlayson-Pitts, 2019).
Likewise, $HO_x$ loss on tropospheric aerosol may significantly impact ozone in high-$NO_x$ atmospheric regimes, as in
East Asia. As particulate levels drop in such regions, $HO_x$ abundance and ozone mixing ratios will both rise (Martin
et al., 2003; Li et al., 2019a; Ivatt et al., 2022). However, reported $HO_2$ uptake coefficients vary widely, from
research group to research group and from the lab to the field (Burkholder et al., 2020), making modeling
predictions highly uncertain.
Both bottom-up and top-down approaches can address chemical complexity. In the traditional bottom-up approach,
the effects on the reaction system of step-by-step additions of chemical complexity are evaluated. This leads to a
better understanding of the fundamental chemistry needed to develop our predictive abilities. Top-down approaches
involve studying chemistry on ambient aerosol particles. This has been done for $N_2O_5$ and $HO_2$ aerosol uptake
(Bertram et al., 2009; Zhou et al., 2021), for heterogeneous OH oxidation (George et al., 2008), and to characterize
SOA formation by using mobile reaction chambers (Jorga et al., 2021). Another top-down method constrains the
rates of multiphase chemistry using detailed, simultaneous measurements of gas-phase composition under a range of
environmental conditions (Brown et al., 2006). Combining top-down and bottom-up approaches enhances our
understanding of the fundamental science while ensuring that parameterizations for atmospheric modeling are
accurate.
Models working over a wide range of spatial and temporal scales can help address issues in chemical complexity. As
mentioned in the previous section, molecular dynamics calculations are becoming increasingly sophisticated. So too
are multiphase kinetics models that can incorporate insights gained at the molecular level into modelling
frameworks that aim to couple the gas phase and condensed phases, including bulk reaction kinetics, mass transfer
and interfacial processes (Poschl et al., 2007; Tilgner et al., 2013; Woo and McNeill, 2015). A challenge is to couple
bulk and interfacial processes correctly. As computing capabilities grow, the complexity of the multiphase and
detailed molecular mechanism models that can be incorporated into chemical transport models will also increase.
Also, Lagrangian-type models increasingly can model specific field observations (Zaveri et al., 2010). We note that
a successful hierarchical approach has arisen in the indoor chemistry community where modeling groups using a
wide range of tools, from molecular dynamics to large-scale computational fluid dynamics, interact closely with
each other and with experimental scientists (Shiraiwa et al., 2019).
**7 Concluding thoughts**

- Multiphase chemistry has evolved alongside the wider field of atmospheric chemistry. While initial studies
  focused on its impacts on the gas phase, the field now addresses how chemistry affects particles. Although
  modification of aerosol composition has direct relevance to climate and human health, we should not lose sight
  of the connection of multiphase chemistry to the gas phase composition of the atmosphere.
- We need to understand chemical processes at the molecular level to improve our ability to interpret field
  observations and predict the nature of a changing atmosphere. Reinforcing an approach based on physico-
  chemical understanding is necessary for detailed predictions of environmental change.
- There are significant research opportunities for the characterization of the chemistry that occurs at the interface
  of the atmosphere with the rest of the environment, such as studies of ocean-atmosphere interactions, indoor air,
  aerosol health effects, atmosphere-cryosphere chemistry from the stratosphere all the way to the snowpack, and
  pathogen-air interactions.
- With increasingly sophisticated experimental and theoretical tools, atmospheric chemical complexity becomes
  more apparent. While exciting, this presents challenges and constraints. We should emphasize not only highly
  detailed, molecular-level measurements but also more widespread and more prolonged aerosol characterization
  that has less chemical specificity but nevertheless provides valuable insights; there is also a role for both remote
  sensing measurements and classical analytical techniques in this regard. This is akin to a need to understand and
  quantify thermal gas phase reactions while also understanding and quantifying microcanonical reactivity.
- Measurements of many fundamental physico-chemical parameters such as solubility, diffusion coefficients, and
  liquid/solid phase reactivities are sorely needed.
- Multiphase chemistry studies conducted under conditions that match those in the atmosphere, including those of
  the free troposphere and lower stratosphere, are needed.
- Using the atmosphere as a laboratory to quantify rates of multiphase processes holds promise, with
  simultaneous measurements of many chemicals and other external parameters becoming more feasible through
  coordinated field measurements. Designing field studies with an eye toward quantification of the multiphase
  reactions is beneficial.
- The field of atmospheric chemistry is healthiest when there is extensive communication and feedback between
  the fundamental chemistry, modeling, and field observation communities (see Figure 1). To keep the three-
  legged stool balanced and strong, multiphase chemists should interact widely with not only other atmospheric
  scientists, but also scientists in related fields such as meteorology, climate dynamics, ecology, and human
  health. This can be accomplished by participation in conferences and seminars that involve science from
  different legs of the stool, students exploring short-term training opportunities in diverse research groups, and
  collaborative grants that bring together laboratory and theory, modeling, and field measurement scientists.

**Author Contributions**

Both authors contributed to writing the manuscript.
**Competing Interests**
The authors declare they have no conflict of interest.
**Acknowledgements**
Thank you to Hind Al-Abadleh, Len Barrie, and Will Fahy for their comments on the manuscript, and to Zilin Zhou
for crafting the figures.

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
