# Peer review of "Opinion: Atmospheric Multiphase Chemistry: Past, Present,"

_EGUsphere, 2023_

## Referee Comment (RC3)

**Review, Opinion paper by Abbatt and Ravishankara , 'Opinion: Atmospheric Multiphase Chemistry: Past, Present, and Future'**

*General*

This is an 'Opinion' contribution to ACPD/ ACP and, as such, hard to review. More than in a regular paper, an 'Opinion' piece must be intended to be subjective and focus according to what the authors rate important. However, a review could assist a bit to at least a bit widen the scope and make the contribution a bit more equilibrated between different work directions which might have contributed to the topic of the contribution. There is considerable confusion with the use of 'multiphase'. It would be great to clarify that.

All in all I would like to suggest to clear the scope and meaning of 'multiphase' and, possibly address some of the details below.

*Details*

Title / Intro: I was a quite surprised when reading the contribution, whether its topic really "Atmospheric multiphase chemistry' or wouldn't 'Atmospheric multiphase and heterogeneous chemistry' possibly be a better fit ? The authors explain they naming in the introduction and it seems to me that their new suggestion is retiring the term "heterogeneous chemistry". I feel it woul be too much do do this. The authors now change from the nomenclature of Ravishankara, A.R.. (1997). Heterogeneous and Multiphase Chemistry in the Troposphere. Science. 276. 1058-1065. 10.1126/science.276.5315.1058 and update in such a way that 'multiphase chemistry' should now include all processes from 2D- interfacial interactions to classical volume reactions. This is understandable as reacto-diffusive lengths can continuously change and there are not only the water droplet and the solid/air interface as extreme cases. However, I would like to suggest the authors offer more explanations for this use of nomenclature which is different from the common use of 'multiphase' not only in the' late 20th century' but also today.

Line 45ff: Do these reviews only cover what was called 'heterogeneous chemistry' or do they also cover 'multiphase multiphase' ? I feel some reviews should be amended here.

Line 48ff: I do not feel the discussion of the 'three-legged stool' really fits here and would suggest to skip this section. Interplay of lab and field measurements with modeling is a topic different from multiphase chemistry. I would also skip the corresponding remark in the abstract, line 16/17. The topic has been well addressed 10 aears ago in the cited Abbatt et al. (2014).

Line 64ff: Now the term 'multiphase chemistry' is being discussed. I feel the text explains what it can be but there is no sharp and applicable definition. What about these cases:

(a) A reaction of two molecules at a surface A(surf) + B(surf) → Products
(b) A reaction 'at the end of the 20$^{th}$ century' clearly addressed as a heterogeneous reaction A(surf) + B(g) → Products

I am wondering if the authors would like to see that the terms 'heterogeneous reaction' or 'heterogeneous process' are to be retired ?

Also, if the new meaning of 'multiphase' is now mainly, and 'in the language of the late 20$^{th}$ century', 'heterogeneous' – what should those groups do which say they study multiphase chemistry ? Change their laboratories ? This is kind of funny.

In summary, under the present title nobody would expect what is treated in the opinion piece. If 'multiphase' is really intended to be treated, add bulk processes. If only heterogeneous processes are to be treated, make it clear from the start. While it is true that the reacti-diffusive length coveres a wide range of distances, it appears useful to discuss its limiting cases, maybe you want to address them as Multiphase process limiting case I = interfacial process and Multiphase process limiting case II = bulk process.

Line 97: 'Thus,…' – No, that is too much said and an incorrect way of using the term 'catalysing'. Catalysing would mean action through the involvement of a catalyst, cf. https://goldbook.iupac.org/terms/view/C00876. Maybe just exchange 'catalysing' by '… can lead to increased reaction rates through concentration enhancements'.

Line 114: Is there examples of a solvent shell in the gas phase ? If no, skip '(generally)'

Lines 107 -114: There is more distinguishing features. The lower effective quantum yields for photolysis processes in solution against the gas phase quantum yields. There is species which do not exist in the gas phase and can get involved in chemical reactions, think of metal ions. On aspect full ymissing in recent work: The role and impact of acidity, Pye et al. (2020), Tilgner et al.(2021).

Line 146: Maybe not only 'cloud water' chemistry but 'aqueous cloud and aerosol chemistry'. I am missing nitrate photolysis for renoxification, role of TMI, and these early studies could be put in perspective with the more recent ones, some of them with severe problems regarding plausible quantum yields.

Line 159 & 160: I don't think this is really true. Both aspects are still treated a lot.

Line 167 : A few examples for particle chemistry would fit well here.

Line 176 ff: Here would be a good place to add work on bulk-phase organic accretion reactions

Line 193: Put earlier and more recent nitrate photolysis work in perspective, see above.

Line 248/249: I would suggest to add Sasho Gligorovski's work here.

Line 338: Here, it could be mentioned that photosensitizer chemistry has in fact been suggested for surface waters early, Canonica, S.; Hoigné, J. (1995)

Line 343/344: There heve been studies where it is shown that the observed effects have to do with extraordinary strong UV radiation and might become negligible under natural UV irridiation. Cf. Fankhauser et al., *ACS Earth and Space Chemistry* 2019, 1485-1491

Line 354: What about development in laboratory techniques ?

Line 373/374: Modern multiphase chemistry modelling can predict the composition of that filter sample by travelling box approaches. What about that ? This part with observational capabilities could be improved considering multiphase modelling.

Line 383 In this part, there are many specific examples but the modelling part is of a quite limited scope. It would be good to strengthen and also link to the preceeding observational capabilities section. Maybe these examples can be kept and discussed but where is a a section dealing with mechanism development, mechanism reduction, automated mechanism generation ? Assess the state of models. How to improve ? Do we have enough computer capacity - Moore's law still holds. And, then, for what ?

Maybe a few more thoughts: There is again and again re-investigation of certain issues which are well implemented into leading models and everybody could look up what certain processes could do – look at S(VI) production in China – how many claims to explain did we see in the last decade ? I 'd say, 100. Fm NO2 to TMI, the uncatalyzed S(IV) oxidation, photosensitization, nitrate photolysis with artificial enhancement - you can find everything. But: problem solved ? No, they are all not doing the job.

But there is exciting new fields (besides those you mention) : Continental halogen multiphase chemistry. Release of hydrated formaldehyde into the gas phase fm aqueous systems. Hydroperoxide and peroxyl radical chemistry. Role of TMI. Coupling of cloud process to aerosol composition (cf. sulfate as a role model !) Accretion product formation in gas and particle phase. Is there interesting NOx(aq) chemistry ? Condensed phase photochemistry.

There is stuff where we must get better: DMS oxidation, aromatics oxidation, multiphase mechanism kernels beyond C4. Having an up-to-date versatile gas-phase backbone mechanism to which everything couples. Have multiphase chemistry mechanisms with interfacial and bulk parts coupled correctly.

---

## Author Comment (AC1)

**Responses to the reviewers' comments for Abbatt and Ravishankara, *Opinion: Atmospheric Multiphase Chemistry: Past, Present, and Future*, 2023**

Reviewers' comments in black, responses in blue, *revised/new text in red*

**Anonymous Referee 1**

This is a well-written paper summarizing the area of multiphase chemistry and some important remaining challenges. It is difficult to appropriately reference all of the contributions to such a broad topic but overall, they have done a good job at this and the paper will be a valuable resource, especially to those not familiar with this area.

That said, a general comment is that there is some inconsistency in the referencing. In some cases, references are made to the original, foundational papers (e.g. line 77, Hoffmann and Edwards, 1975; Penkett et al., 1979; also references on lines 128, 130). In this reviewer's opinion, this is excellent as there is an increasing tendency in the literature to only cite the last few years work that would not have happened without the seminal papers such as these. Since this paper will undoubtedly be read and highly cited, referencing the early work is important.

Response: Thank you for the very helpful review.  Yes, we hope that the paper will be useful to new researchers in the field.

The reviewer is correct that it is a challenge to put references in this paper.  As stated in the original manuscript, it is not a review paper and the referencing was not intended to be complete (line 44 of original draft: "*For the sake of brevity, the citations in this paper are illustrative and not comprehensive.*").  We have now edited the paper to make it clear that this paper was solicited for the Special Issue entitled "20 Years of Atmospheric Chemistry and Physics" whose goal is to "to reflect on the developments of the field of atmospheric chemistry and physics in the last 20 years and point to exciting directions for the future".  This is the reason for the preponderance of references from the 2000's.

The additional text is (line 36):

*"As part of the Special Issue entitled "20 Years of Atmospheric Chemistry and Physics" both authors value the current opportunity to contribute to the overall goal of the special issue "to reflect on the developments of the field of atmospheric chemistry and physics in the last 20 years and point to exciting directions for the future" by addressing the evolution of the field of atmospheric multiphase chemistry."*

That all said, we fully agree that it is important to place new work in the context of foundational studies.  Thanks for the suggestions!  We have incorporated all the referees' reference suggestions and we added additional ones of our own.  For example, the section on early advances in the field has been re-written with additional pioneering references included (line 157):

*"The idea of reactive chemistry in hydrometeors goes back to the late 1960s and 1970s and possibly earlier. As in much of science, it is hard to pinpoint a specific paper that expounded this idea. The expectation that the $SO_2$ pollutant can be oxidized to sulfuric acid in the water droplets in the atmosphere spurred many studies, hypotheses, and definitive results. Early studies examined the potential oxidation of $SO_2$ in the liquid phase via a variety of oxidants, including $O_3$, $H_2O_2$, and $NO_2$ (Hoffmann and Edwards, 1975; Erikson et al., 1977; Schroeder and Urone, 1978; Penkett et al., 1979). As well, modeling studies showed the feasibility of such oxidation reactions occurring in the atmosphere (Jacob and Hoffmann,*

*1983; and references therein). The Great Dun Fell experiment observationally established that SO₂ is indeed oxidized in the troposphere via reactions in cloud droplets (Choularton et al., 1997).*

*Similar multiphase reactions in the stratosphere were sometimes evoked but never pursued with great vigor until the spectacular occurrence and subsequent explanation of the ozone hole (Solomon, 1999) and a multitude of laboratory studies showing that indeed there can be chlorine activation (Molina et al., 1987; Tolbert et al., 1988; Leu, 1988; Hanson and Ravishankara, 1992; and many others)."*

On the other hand, there are a number of instances where only recent papers are cited, and references to the earlier work would be helpful to a reader, e.g.,

Line 137: While Carpenter et al. (2013) have done very nice work on the ozone-iodide reaction, there are foundational papers on this that go back many decades before this that would be helpful to cite.

Response: Agreed. The reviewer may have missed references in the original manuscript (Line 135) to the work of Miyake and Tsunogai, 1963 and Garland et al. 1980.

Line 167: Pankow et al. published many papers on partitioning and volatility starting in the 1980's, which laid the foundation for the Donahue et al. subsequent studies, and should be cited.

Response: Agreed. We have added: Pankow, 1994.

Line 206: Optical properties and photodegradation of brown carbon are discussed in detail in a 2015 Chem Rev article by Laskin et al and should be cited in addition to Hems et al.

Response: Agreed. The Laskin et al. reference is now included at this location.

Line 249: Gligorovski et al published a number of papers on indoor OH (e.g. Atmos. Env., 2014) and indoor chemistry, long before the Sloan Foundation funding pushed this area to the fore. Some of his work should be cited.

Response: The intent of this sentence is to relate the multiphase chemistry occurring between ozone and skin oils to gas phase OH generation via ozonolysis in indoor environments. That said, we have added one publication from Gligorovski and co-workers where indoor OH is attributed to photolysis of HONO, which is likely formed to some degree via multiphase chemistry on indoor surfaces. The new text is (line 302):

*"This source of OH from alkene ozonolysis is in addition to the generation of OH from photolysis of HONO (Gomez Alvarez et al., 2013), which is partly formed by multiphase chemistry on indoor surfaces."*

Line 267: Some of the original work on particles translocating to the brain, e.g. Oberdorster et al., should be cited.

Response: Agreed. We have added: Oberdorster et al., 2004. We have also added Maher et al., 2016.

Line 306: Zelenyuk et al field measurements on shielding of PAH inside particles should also be cited.

Response: Agreed. We have added: Shrivastava et al., 2017.

**Other reference suggestions:**

Line 161: Citing broader treatments of health effects of particles would be appropriate, e.g. some of the Landrigan et al articles summarizing the results from the commission on pollution and health.

Response: Agreed. We have added Landrigan et al., 2018.

Lines 351-353: Adding some references supporting the statements about absorption spectra and quantum yields being different from those in the gas phase would be helpful to the reader. Also mentioning the effects of viscosity and matrix on these (e.g. Nizkorodov et al)

Response: Thank you. Yes, the gas phase species taken up by a substrate such as ice or snow can change the character of the species and hence its photochemistry. For potential impacts on the atmosphere, we now refer to a review from 2002 of Domine and Shepson that highlights some of the processes. More specific examples from field work in Greenland and the Antarctic (Honrath et al., 1999; Wolff et al., 2002) were already cited in the paper. We have revised the text for this section, including a review article for atmospheric condensed phase photochemistry (George et al., 2015) and a specific recent reference for nitrate aqueous photochemistry (Benedict et al., 2017). To address the impact of viscosity, we have added one reference (Lignell et al., 2014):

The revised text is (line 412):

*"In addition to indirect sensitization, light can also lead to direct photochemistry. An important finding was that photolysis on ice and snow surfaces was demonstrated to form $NO_x$ in midlatitudes and polar regions (Honrath et al., 1999; Wolff et al., 2002; Domine and Shepson, 2002). This process, which likely proceeds in a wide range of environments, is now referred to as "re-noxification" as it releases $NO_x$ from $HNO_3$ that has deposited from the atmosphere. Other condensed-phase chemical processes of importance include the formation of oxidants from nitrate and nitrite photolysis (Zepp et al., 1987), photolysis of condensed-phase organic hydroperoxides and other highly oxygenated organics, and the photochemical activity of many transition metal ion complexes (Faust and Zepp, 1993; Weller et al., 2013). It is important to recognize that the absorption spectra and product quantum yields of dissolved species can be different than those in the gas phase (George et al., 2015), with aqueous nitrate a prime example (Benedict et al., 2017). The variable viscosity of organic aerosol matrices can affect photolysis rates, products, and their temperature dependence (Lignell et al., 2014)."*

Line 359: Zelenyuk and their SPLAT should be included here.

Response: Agreed. We have added: Zelenyuk and Imre, 2005.

**Other comments:**

Line 180: The impact of temperature on SOA formation is mentioned. However, growth of particles is also affected and should be mentioned.

Response: Agreed. The next text reads (line 229):

*"Organic particles are likely glasses in the cold free troposphere (Shiraiwa et al., 2017), which may affect SOA formation and growth, and restricts the degree to which heterogeneous oxidation can affect the aerosol composition."*

Lines 211, 212: A brief description of what is meant by the deposition and immersion modes would be helpful to non-experts in the field.

Response: Agreed. The new text reads (line 260):

*"Multiphase chemistry can also affect the properties of ice nucleating particles (INPs) by both gas-solid and liquid-solid interactions, noting that INPs can act in the deposition mode where water vapor forms ice directly on solid substrates and in the immersion mode where a solid particle immersed in supercooled water leads to nucleation (Kanji et al., 2017)."*

Lines 256-259: I think this statement regarding the impacts on health is grossly understated. Particles have not been "implicated" as harmful, there is a ton of evidence showing they are. The statement that the toxicity mechanisms are not known is also somewhat misleading. For example, it has been known for decades that particles initiate an inflammatory response.

Response: Good comment. We have adjusted the text accordingly and have also indicated that uncertainty in the toxic species and toxicity mechanism remains at the molecular level (line 310):

*"Epidemiological studies have conclusively shown that aerosol particle inhalation harms human health (Pope et al., 2009; Landrigan et al., 2018). For example, it is well-recognized that inflammation occurs upon particle exposure (Brook et al., 2003). The current epidemiology (empirical evidence) does not readily distinguish the specific molecules in the particles and their formation pathways that lead to negative health outcomes, nor the toxicity mechanism at the molecular level. Studies are currently examining oxidative stress, e.g., the reactive oxygen species (ROS) and reactive nitrogen species (RNS), as a mechanism for negative impacts (Shiraiwa et al., 2017a). Although there is debate over whether oxidants are largely endogenous or exogenous (Fang et al., 2022), one hypothesis is that the biochemical balance between oxidants and antioxidants is upset by inhaling harmful species (Miller, 2020)."*

Line 291: The last sentence "It is not clear whether this effect is related to multiphase chemistry at biological surfaces" seems out of place. What is meant by this needs to be amplified or the sentence omitted.

Response: Our intent here was to suggest that multiphase chemistry may occur between oxidants in the air and the wound surface. We have re-written the text to make this clearer (line 349):

*"It is crucial to determine whether these effects are related in some way to multiphase chemistry occurring between reactive species in the gas phase interacting with biological molecules at the surface of the wounds and lungs."*

Lines 309-311: The authors might want to mention that freezing also concentrates soluble species in the shrinking liquid layer, which can have significant impacts on the chemistry.

Response: Good suggestion. Revised text (line 369):

*"In addition to continuing to address the fundamentals of cloud chemistry oxidation processes, the associated chemistry of transition metals, and the production of oxidants within cloud water and via uptake from the gas phase (Herrmann et al., 2015), there is a particular need to study such processes at cold temperatures, especially under supercooled water conditions. When supercooled water is frozen, solutes are excluded from the ice crystals and become highly concentrated at grain boundaries and in liquid and liquid-like layers at the surface, potentially leading to enhanced rates of aqueous phase*

*chemistry. As well, the Reynolds-Workman potential (Workman and Reynolds, 1950), arising at the ice-water interface, can drive chemistry."*

Indeed, one comment we received on the paper was that we should emphasize the interactions with the cryosphere in the final section, so we have modified this comment (line 532):

"*There are significant research opportunities for the characterization of the chemistry that occurs at the interface of the atmosphere with the rest of the environment, such as studies of ocean-atmosphere interactions, indoor air, aerosol health effects, atmosphere-cryosphere chemistry from the stratosphere all the way to the snowpack, and pathogen-air interactions.*"

Section on chemical complexity: This is a nicely written section. In the complexity listed on lines 402-403, the authors might also want to mention the importance of the surface composition vs the bulk, e.g. Wingen et al, Chem Sci (2019).

Response: We agree that this is an issue that leads to considerable chemical complexity.  This sentence now reads (line 494):

*"This complexity is exacerbated by the changes in the composition (including acidity), mixing state, and water content of the particle as it resides in the atmosphere. Added complexity arises from the differences in composition that occur in the bulk of particles and droplets compared to their interfacial composition (Wingen and Finlayson-Pitts, 2019)."*

**Anonymous Referee 2**

This is a terrific, forward-looking review on multiphase chemistry written by two giants of the field. I believe that it will be widely read and cited. It is destined to be a wonderful resource for students and experts alike. My thanks to the authors for writing it.

Response: Thank you for the very helpful review. Yes, we hope that the paper will be useful.

Here are a few comments that I encourage the authors to consider before final publication:

70: The definition of the "boundary" appears to leave out its role as a thin region in which reactions as well as transport can occur. This possibility is mentioned explicitly in the next paragraph.

Response: Agreed. We have significantly increased the wording describing the interface in Section 2. The relevant sentence now reads (line 85):

*"The interface is involved in all multiphase processes, participating in mass transfer and reactivity."*

I like figure 1, but the authors may wish to add a dashed line from Csurf pointing downward toward the bulk in the left hand panel (and all three panels): even if the reaction occurs in the interfacial region because A reacts quickly, the product C may be soluble and diffuse widely, or it may even be ionic and stay within solution.As a technical point, the reacto-diffusive length may not really apply to the left hand panel, at least as it is customarily defined by sqrt(D/k), where D and k are bulk-phase quantities. D and k in the interfacial region might not be similar to their bulk-phase values. It's not just that the bulk-phase reaction region is shallow, but that the energetics and solvation and motions could be quite different in the interfacial and bulk-phase regions.

Response: Agreed. We have adapted the figure as suggested, where $C_{surf}$ can diffuse downwards into the bulk in the middle and right frames of the figure. We have not made this change for the left frame which represents a system with long diffusion times.

To address the second point, we have added the following text to the paper (line 98):

*"It is not easy to determine whether a reaction is occurring exclusively in a two-dimensional manner at an interface, where the concept of reacto-diffusive depth breaks down because of different energetics and solvation than in the bulk."*

and to the figure caption (line 112):

*"As noted in the text, the concept of the reacto-diffusive length, which is formally calculated from the bulk diffusion and rate constants, may break down in the interface region."*

91: I recommend adding "neutral" to "closed shell". Reactions of charged species can be fast in the gas phase.

Response: Agreed. The text has been adjusted as recommended.

101: does "sorbed" mean at the surface and in the subsurface region?

Response: Yes, the intention of using the word "sorbed" was to capture both the interfacial and dissolved solute. To be more specific, we have adjusted the text to (line 124):

*"For example, it is likely that both adsorbed and dissolved HCl dissociates on/in ice to form chloride ions, which are reactive with $ClONO_2$."*

102: Is there evidence that Cl+ forms as an independent solvated species? Perhaps it is more like an SN2 reaction with Cl- acting as a nucleophile attacking ClONO2. This idea follows from work by the J. T. Hynes group and others of this reaction on ice surfaces.

Response: We agree that there is no direct experimental evidence for $Cl^+$ as an independently solvated species. In the gas phase, we note that the energetics for formation of different product pairs from dissociation of $ClONO_2$, i.e., either $Cl^-/NO_3^+$ or $Cl^+/NO_3^-$, indicates that there is no preference for one set of products over the other. We do not know these values in solution. Yet, it appears possible that $ClONO_2$ dissociates to give $Cl^+ + NO_3^-$ because we do see formation of $Cl_2$ in that reaction. In other words, the reactions of $ClONO_2$ with HCl is consistent with this picture for the formation of $Cl_2$ (via reaction between $Cl^+$ and $Cl^-$). Alternatively, Hynes and coworkers have suggested that a reaction between $Cl^-$ and $ClONO_2$ to give $Cl_2$ and $NO_3^-$ is possible, where there is a partial positive charge ($\delta+$) on the Cl atom in $ClONO_2$. To the best of our knowledge, there are no experimental evidence to show the occurrence of an SN2 type of reaction. As the reviewer notes, the products would be the same. We have made the following changes to the manuscript (line 112):

*"For example, it is likely that both adsorbed and dissolved HCl dissociates on/in ice to form chloride ions, which are reactive with $ClONO_2$:*

$$ClONO_2 + HCl \rightarrow Cl_2 + HNO_3 \qquad R2$$

*leading to the formation of $Cl_2$, which is then released to the gas phase and is readily photolyzed, forming radicals that catalyze gas phase ozone destruction in the Antarctic ozone hole. Also, $ClONO_2$ can be protonated in strongly acidic cloud droplets, or it may dissociate to form $Cl^+$ and $NO_3^-$ (although there is no direct experimental evidence yet for forming $Cl^+$ as an independently solvated species). Other possible mechanisms include a concerted reaction between $Cl^-$ and $ClONO_2$ to produce $Cl_2$ (Bianco and Hynes, 1999). It is unclear if such a reaction is truly an SN2-type process or if it can occur in a cage in the condensed phase."*

402: Is "exasperated" the correct word? Perhaps "exacerbated" or "amplified"?

Response: The wording has been changed to "exacerbated". (Perhaps the authors were exasperated at something or other when they wrote that text! We could also blame the autocorrect feature! But it now fixed, thank you.).

426: I think that "the" in "the theory" should be deleted.

Response: We have reworded this sentence.

419 - 430. I would like to recommend citations for two pioneers and their groups in the theoretical community for deducing multiphase reaction mechanisms. They are not mentioned here: J. T. Hynes and R. B. Gerber. They could be cited through their 2006 and 2015 Accounts of Chemical Research articles.

Response: Excellent suggestion. Our apologies for overlooking this pioneering work. We have re-written the relevant paragraph to include these example (line 462):

*"Molecular-level chemical models provide valuable insights to complex multiphase processes. For example, important insights into the nature of the chemistry occurring on polar stratospheric cloud materials were obtained from molecular dynamics modeling (Wang and Clary, 1996; Bianco and Hynes, 2006) and more recent studies have addressed gas-surface interactions and the roles of solvent molecules in small molecular clusters (Gerber et al., 2015; Fang et al., 2019; Yang et al., 2019). Whereas past computational methods only included a few solvent molecules, current dynamics models using state-of-the-art force fields can realistically simulate partitioning, surface adsorption constants, diffusion constants, and vapor pressures, representing an important point of contact to the physical chemistry and chemical physics communities (Tobias et al., 2013). We also note that machine-learning techniques have been applied to molecular dynamics simulations to describe the interactions of $N_2O_5$ with liquid water (Galib and Limmer, 2021) and the dissociation of strong acids at aqueous interfaces (de la Puente et al., 2022). For establishing fundamental parameters that are experimentally challenging to measure, such as the likelihood that a collision of a molecule with a particle leads to uptake by the condensed phase (i.e., a mass accommodation coefficient), theoretical methods may be preferable to experiment in some situations."*

lines 438-9 about AI and machine learning.

I think that this statement may be too sweeping. AI could be used as a high-dimensional data fitting tool that makes predictions but does not necessarily reveal new insights. This may be what the authors have in mind. But machine learning can also be used to build molecular models that lead to new and powerful mechanistic insights. Two examples of the latter are J. Am. Chem. Soc. 2022, 144, 10524−10529 Acids at the Edge: Why Nitric and Formic Acid Dissociations at Air−Water Interfaces Depend on Depth and on Interface Specific Area Science 371, 921–925 (2021) Reactive uptake of N2O5 by atmospheric aerosol is dominated by interfacial processes. I hope that the authors can mention these new directions, as machine learning may become an essential tool to partner with laboratory experiments.

Response: Agreed. We have deleted that specific sentence about AI and we have included these two examples about machine learning in the new paragraph described in the previous response.

459 - 462. Nicely stated. How will the authors motivate the multiphase community to interact more closely with other atmospheric chemists? As leaders of the field, the authors can benefit the community by making specific recommendations and leading us in these recommendations.

Response: We have added a number of suggestions to facilitate such interactions (line 550):

*"The field of atmospheric chemistry is healthiest when there is extensive communication and feedback between the fundamental chemistry, modeling, and field observation communities (see Figure 1). To keep the three-legged stool balanced and strong, multiphase chemists should interact widely with not only other atmospheric chemists, but also scientists from related fields such as meteorology, climate dynamics, ecology, and human health. This can be accomplished by participation in conferences that involve science from different legs of the stool, students exploring short-term training opportunities in diverse research groups, and collaborative grants that bring together laboratory and theory, modeling, and field measurement scientists."*

**Review by Hartmut Herrmann**

*General*

This is an 'Opinion' contribution to ACPD/ ACP and, as such, hard to review. More than in a regular paper, an 'Opinion' piece must be intended to be subjective and focus according to what the authors rate important. However, a review could assist a bit to at least a bit widen the scope and make the contribution a bit more equilibrated between different work directions which might have contributed to the topic of the contribution. There is considerable confusion with the use of 'multiphase'. It would be great to clarify that.  All in all I would like to suggest to clear the scope and meaning of 'multiphase' and, possibly address some of the details below.

Response:  We thank Dr. Herrmann for the very helpful review.  The issues raised in this review prompted us to broaden our discussion and to sharpen some concepts within it, in particular with respect to the use of the term "multiphase".

Based on the reviews, we felt it necessary to clarify that this is not a review article but rather an opinion article solicited for the ACP Special Issue entitled "*20 Years of Atmospheric Chemistry and Physics*" whose stated goal is to "*to reflect on the developments of the field of atmospheric chemistry and physics in the last 20 years and point to exciting directions for the future*".

The additional text is (line 36):

*"As part of the Special Issue entitled "20 Years of Atmospheric Chemistry and Physics" both authors value the current opportunity to contribute to the overall goal of the special issue "to reflect on the developments of the field of atmospheric chemistry and physics in the last 20 years and point to exciting directions for the future" by addressing the evolution of the field of atmospheric multiphase chemistry."*

*Details*
Title / Intro: I was a quite surprised when reading the contribution, whether its topic really "Atmospheric multiphase chemistry' or wouldn't 'Atmospheric multiphase and heterogeneous chemistry' possibly be a better fit ? The authors explain they naming in the introduction and it seems to me that their new suggestion is retiring the term "heterogeneous chemistry". I feel it woul be too much do do this. The authors now change from the nomenclature of Ravishankara, A.R.. (1997). Heterogeneous and Multiphase Chemistry in the Troposphere. Science. 276. 1058-1065. 10.1126/science.276.5315.1058 and update in such a way that 'multiphase chemistry' should now include all processes from 2D- interfacial interactions to classical volume reactions. This is understandable as reacto-diffusive lengths can continuously change and there are not only the water droplet and the solid/air interface as extreme cases. However, I would like to suggest the authors offer more explanations for this use of nomenclature which is different from the common use of 'multiphase' not only in the' late 20th century' but also today.

Response:  This is a very important point, i.e., delineating the utility and meaning behind the terms "multiphase chemistry" and "heterogeneous chemistry".  As Dr. Herrmann mentions, in the review paper referred to above (Ravishankara, 1997), the term "heterogeneous chemistry" was used to refer to gas-solid reactions whereas the term "multiphase chemistry" referred to reactions in liquids where bulk processes are prevalent in addition to interfacial processes.  The term "heterogeneous chemistry" likely arises from the use of "heterogeneous catalysis" in the chemical catalysis and kinetics literature, which refers to chemistry occurring at an interface between phases (Svehla, 1993).

As mentioned in the original version of the manuscript (lines 40 to 41), the atmospheric chemistry field has evolved to recognize that there is a continuum of bulk diffusivities such that there are condensed

phases that act as true liquids under some environmental conditions and as semi-solids or glasses under other conditions. Even for substrates that are pure solids such as mineral dusts, we now know that they have variable amounts of adsorbed water and are frequently coated with organics and sulfate. As such, there is a blurring in the distinction between the two endpoints represented by reactions that occur only at the interface of a solid or entirely within the bulk portion of a liquid droplet.

It is the authors' opinion that the term "multiphase chemistry" encompasses this continuum of reaction scenarios/depths. Specifically, the term implies chemistry occurring in more than one phase, and interfaces are invariably present when multiple phases are in contact with each other. We note that a useful parameter for separating heterogeneous processes from non-heterogeneous processes is the gradient in density across an interface. For a pure solid, where heterogeneous chemistry can occur, this gradient is best represented by a step function. However, the interfacial region for coated mineral dust, a salt solution particle, or an organic particle might be a few molecules thick at least, or else much deeper.

It is our impression that the field is moving in the direction of referring to aerosol and droplet chemistry as "multiphase chemistry", motivating the use of this term throughout the paper. However, we stress that the term "heterogeneous chemistry" is still useful to refer to exclusively interfacial processes; we did not intend to suggest that it should be phased out. Likewise, the term "bulk chemistry", as may occur in the bulk component of large water droplets, is also a useful expression.

These comments were extremely valuable, and they prompted us to add an extensive discussion on this and related topics in two locations in the revised version of the manuscript (line 47):

*"In the 1997 paper, Ravishankara distinguished between heterogeneous and multiphase chemistry based on the extent of diffusion into the bulk. At that time, the term "heterogeneous chemistry" was in vogue to describe ozone hole chemistry. Over the years, it has become clear that diffusion depths vary continuously from solid-like substrates to dilute water solutions. Therefore, in this article, we use the term "multiphase chemistry" to refer to all chemistry that involves more than one phase. Interfacial chemistry falls under this umbrella, with interfaces invariably present when more than one phase is present. We note that "heterogeneous chemistry" is a useful term to describe exclusively interfacial processes (Svehla, 1993), such as for the reactions of gas phase molecules and atoms on solid material such as metallic or mineral catalysts. Similarly, "bulk chemistry" refers to chemistry that occurs mainly in only one phase. In this article, our focus is primarily on processes involving the gas phase interacting with atmospheric condensed phases, so we do not describe in-depth advances in the associated chemistry that takes place in the bulk phase."*

and (line 85)

*"The interface is involved in all multiphase processes, participating in mass transfer and reactivity. For a low-volatility material, such as a bare mineral or metal, there is a step function drop in the density moving from the condensed phase to the gas phase. A solid's interface, roughly one molecule (or atom) deep, can promote heterogeneous chemistry that occurs in a two-dimensional space (Tabazadeh and Turbo, 1993). However, solid particles in the atmosphere, such as soot or mineral dust, frequently have mobile, semi-volatile molecules, such as water, sulfuric acid, and organics, deposited as multilayer-thick films or islands. In these cases, the multiphase chemistry is likely occurring in a three-dimensional space involving chemical interactions with not only the solid substrate but also the liquid coatings (Worsnop et al., 2002; Poschl et al., 2007). For cloud droplets and liquid aerosol particles, which are very dynamic with large fluxes of substrate molecules being rapidly exchanged between the phases, the interfacial region is also thicker than it is for a pure solid (Gopalakrishnan et al., 2005). When the underlying substrate is a liquid, the depth to which a molecule diffuses from the interface into the bulk before reacting is called the reacto-diffusive length, which can be very short for species reacting close to the*

*interface and extending to the size of an aerosol particle for reactions that are aerosol-volume limited (see Figure 2) (Hanson et al., 1994). It is not easy to determine whether a reaction is occurring exclusively in a two-dimensional manner at an interface, where the concept of reacto-diffusive depth breaks down because of different energetics and solvation than in the bulk phase. For particle chemistry, the reaction kinetics dependence on particle size and reactant concentration can provide some information. Nevertheless, even in reactions that exhibit Langmuir-Hinshelwood kinetics (Poschl et al., 2001), the depth of the reactive region at the surface of a particle is not clear for liquids or coated-solids."*

Line 45ff: Do these reviews only cover what was called 'heterogeneous chemistry' or do they also cover 'multiphase multiphase' ? I feel some reviews should be amended here.

Response: We have added the following reviews on aqueous chemistry and particle/droplet acidity (McNeill, 2012; Rudich et. al., 2007; Herrmann et al., 2015; McNeill, 2015, Pye et al., 2020; Tilgner et al., 2021).

Line 48ff: I do not feel the discussion of the 'three-legged stool' really fits here and would suggest to skip this section. Interplay of lab and field measurements with modeling is a topic different from multiphase chemistry. I would also skip the corresponding remark in the abstract, line 16/17. The topic has been well addressed 10 aears ago in the cited Abbatt et al. (2014).

Response: Thanks for this discussion but we have decided to leave this discussion at its original location, at the end of the Introduction. We think it is valuable to alert the reader to themes that the authors view as being important, i.e., as a "heads-up" before delving into the heart of the paper.

Line 64ff: Now the term 'multiphase chemistry' is being discussed. I feel the text explains what it can be but there is no sharp and applicable definition. What about these cases:
(a) A reaction of two molecules at a surface A(surf) + B(surf) → Products
(b) A reaction 'at the end of the 20th century' clearly addressed as a heterogeneous reaction A(surf) + B(g) → Products
I am wondering if the authors would like to see that the terms 'heterogeneous reaction' or 'heterogeneous process' are to be retired ? Also, if the new meaning of 'multiphase' is now mainly, and 'in the language of the late 20th century', 'heterogeneous' – what should those groups do which say they study multiphase chemistry ? Change their laboratories ? This is kind of funny. In summary, under the present title nobody would expect what is treated in the opinion piece. If 'multiphase' is really intended to be treated, add bulk processes. If only heterogeneous processes are to be treated, make it clear from the start. While it is true that the reacti-diffusive length coveres a wide range of distances, it appears useful to discuss its limiting cases, maybe you want to address them as Multiphase process limiting case I = interfacial process and Multiphase process limiting case II = bulk process.

Response: Please see the response above concerning this topic. As mentioned, we do not propose that the term "heterogeneous chemistry" should no longer be used.

Line 97: 'Thus,…' – No, that is too much said and an incorrect way of using the term 'catalysing'. Catalysing would mean action through the involvement of a catalyst, cf. https://goldbook.iupac.org/terms/view/C00876. Maybe just exchange 'catalysing' by '… can lead to increased reaction rates through concentration enhancements'.

Response: Good point. We have deleted this sentence.

Line 114: Is there examples of a solvent shell in the gas phase ? If no, skip '(generally)'

Response:  We had put the term "(generally)" into the paper to refer to gas phase reaction systems where a chaperone can play a role, e.g. the role of $H_2O$ molecules in the gas-phase self-reaction of $HO_2$. We have simplified the text (line 141):

*"In the gas phase, the solvent shell is essentially absent (except for some specific cases, such as cluster formation with $H_2O$), so the carbonyl and Criegee intermediates fly apart, and secondary ozonides do not form so readily."*

Lines 107 -114: There is more distinguishing features. The lower effective quantum yields for photolysis processes in solution against the gas phase quantum yields. There is species which do not exist in the gas phase and can get involved in chemical reactions, think of metal ions. On aspect full ymissing in recent work: The role and impact of acidity, Pye et al. (2020), Tilgner et al.(2021).

Response:  Good points. The different photolysis yields are mentioned in Section 6.5. And, it is true, in the original manuscript we did not emphasize the role of transition metal ions as much as we should have. They are now mentioned at a number of locations in the paper, along with new text at this location (line 144):

*"Lastly, some multiphase reactants, such as transition metal ions, are not present in the gas phase. More generally, ion-ion and ion-molecule regions play a much greater role in condensed phase chemistry than in tropospheric and stratospheric gas phase chemistry, leading to a wide variety of novel reaction pathways, with R1 being an excellent example."*

We have added these references, as well as McNeill et al. (2012), Herrmann et al. (2015), McNeill (2015).

Line 146: Maybe not only 'cloud water' chemistry but 'aqueous cloud and aerosol chemistry'. I am missing nitrate photolysis for renoxification, role of TMI, and these early studies could be put in perspective with the more recent ones, some of them with severe problems regarding plausible quantum yields.

Response: Thanks. The wording has been clarified here. New text (line 188):

*"iv) the impact of aqueous cloud and aerosol chemistry on gas phase $HO_x$ levels (Chameides and Davis, 1982; Calvert et al., 1985; Jacob, 1986; Lelieveld and Crutzen, 1991)"*

The focus of this paragraph was largely on gas-aerosol and gas-droplet processes studied before 2000; we now clarify this in the text.

However, the topics raised are important and are now addressed through the following new text (line 197):

*"Additional work in the multiphase world at this time involved a wide variety of condensed phase photochemistry studies, for example involving the interactions of light with nitrate (Zepp et al., 1987), which can lead to the release of $NO_x$ to the gas phase, and with transition metal ion complexes (Faust and Zepp, 1993)."*

We note that the role of TMI is now addressed at a number of new locations in the text (lines 252, 369, 416):

*"As well, the optical properties of the fraction of organic aerosol that absorbs near ultraviolet and visible light (i.e., atmospheric 'brown carbon' particles) is subject to change via multiphase oxidation and condensed phase photochemistry (Laskin et al., 2015; Li et al., 2020a; Hems et al., 2021; Schnitzler et al., 2022), potentially involving transition metals (Al-Abadleh and Nizkorodov, 2021)."*

*"In addition to continuing to address the fundamentals of cloud chemistry oxidation processes, the associated chemistry of transition metals, and the production of oxidants within cloudwater and via uptake from the gas phase (Herrmann et al., 2015), there is a particular need to study such processes at cold temperatures, especially under supercooled water conditions."*

*"Other condensed phase photochemical processes of importance include the formation of oxidants from nitrate and nitrite photolysis (Zepp et al., 1987), photolysis of condensed phase organic hydroperoxides and other highly oxygenated organics, and the photochemical activity of many transition metal ion complexes (Faust and Zepp, 1993; Weller et al., 2013)."*

Line 159 & 160: I don't think this is really true. Both aspects are still treated a lot.

Response:  This comment pertains to the emphasis in the field on the role of "particles affecting the gas phase composition" versus "the gas phase affecting the particle composition".   It is the authors' impression that there was a shift in emphasis from the former to the latter at roughly this time, two decades ago. However, we agree that both aspects are still being addressed and the original wording may have been too strong. So, this statement has been reworded (line 205):

*"As a result, additional research emphasis in the multiphase chemistry community was given at the turn of the 21$^{st}$ century to assess impacts that arise on the composition of the particles."*

Line 167 : A few examples for particle chemistry would fit well here.
Line 176 ff: Here would be a good place to add work on bulk-phase organic accretion reactions

Response:  We agree. This is a very large topic, so only a few illustrative references are included (line 214):

*"Specifically, volatility and multiphase reactivity can be coupled, as illustrated by the formation of high molecular weight, low volatility species within particles (Kalberer et al., 2004).  While such oligomers and highly oxygenated species may also form in the gas phase (Bianchi et al., 2019), they arise via a variety of reactions involving pairs of organic reactants, frequently forming esters and acetals/hemiacetals in the condensed phase (Tobias and Ziemann, 2000; Surratt et al., 2006; DeVault and Ziemann, 2021). These reactions may be acid-catalyzed (Jang et al., 2002)."*

Line 193: Put earlier and more recent nitrate photolysis work in perspective, see above.

Response:  We have directed the reader to the section of the paper that mentions nitrate photolysis. New text (line 241):

*" … formation of condensed-phase oxidants through the photolysis of particulate nitrate (Zheng et al., 2020; see also Section 6.5) …"*

and we have re-written that section of the paper on condensed phase photochemistry (Section 6.5, line 412)):

*"In addition to indirect sensitization, light can also lead to direct photochemistry. An important finding was that photolysis on ice and snow surfaces was demonstrated to form $NO_x$ in midlatitudes and polar regions (Honrath et al., 1999; Wolff et al., 2002; Domine and Shepson, 2002). This process, which likely proceeds in a wide range of environments, is now referred to as "re-noxification" as it releases $NO_x$ from $HNO_3$ that has deposited from the atmosphere. Other condensed-phase chemical processes of importance include the formation of oxidants from nitrate and nitrite photolysis (Zepp et al., 1987), photolysis of condensed-phase organic hydroperoxides and other highly oxygenated organics, and the photochemical activity of many transition metal ion complexes (Faust and Zepp, 1993; Weller et al., 2013). It is important to recognize that the absorption spectra and product quantum yields of dissolved species can be different than those in the gas phase (George et al., 2015), with aqueous nitrate a prime example (Benedict et al., 2017). The variable viscosity of organic aerosol matrices can affect photolysis rates, products, and their temperature dependence (Lignell et al., 2014)."*

Line 248/249: I would suggest to add Sasho Gligorovski's work here.

Response:  The intent of the original sentence was to relate the multiphase ozonolysis chemistry to a source of gas phase OH.  We have extended the discussion by including one publication from Gligorovski and co-workers where indoor OH is attributed to photolysis of HONO, where some fraction of indoor HONO is likely formed via multiphase chemistry. The new text is (line 302):

*"This source of OH from alkene ozonolysis is in addition to the formation of OH from photolysis of HONO (Gomez Alvarez et al., 2013), which is partly formed by multiphase chemistry on indoor surfaces."*

Line 338: Here it could be mentioned that photosensitizer chemistry has in fact been suggested for surface waters early, Canonica, S; Hoigne, J (1995).

Response:  Thanks. This reference has been added at this location.

Line 343/344: There have been studies where it is shown that the observed effects have to do with extraordinary strong UV radiation and might become negligible under natural UV irradiation, cf. Frankhauser et al. ACS Earth and Space Chemistry 2019, 1485-1491

Response: The text at lines 343/344 is addressing atmospheric photosensitizers. However, the (interesting!) paper recommended by the reviewer is on the role of bacteria in cloud water chemistry. Perhaps we are missing the connection but given this mismatch we have not addressed this comment.

Line 354: What about development in laboratory tecnhniques?

Response:  This is a good point to raise. The reason to have a section called "Developments in Observational Capabilities" was to highlight the degree to which new field observations will drive the overall field forward. Of course, laboratory techniques will develop too, ideally in sync with field observations.  We have addressed this comment by including a new section called "Development in Laboratory and Molecular Modeling Techniques" where we have included text on molecular modeling that was originally in section "Section 6.7 Grappling with chemical complexity".  This new section is (line 452):

*"Section 6.7  Developments in Laboratory and Molecular Modeling Techniques*

*In addition to our ability to conduct field observations, a revolution has occurred in the laboratory's analytical methods. This is most widely apparent in applying sophisticated mass spectrometric*

*techniques, increasingly involving high mass resolution and a range of ionization schemes (Laskin et al., 2013). When coupled with other analytical methods, we can now determine the physico-chemical properties of individual molecules and their mixtures in extreme detail. For example, this approach has been taken to characterize the optical properties of brown carbon aerosol materials (Fleming et al., 2020), the viscosity of organic aerosol (DeRieux et al., 2018), and the structural isomers of complex organic mixtures (Krechmer et al., 2016). There are significant opportunities for additional adoption of techniques from neighboring disciplines. As well, the use of a number of these analytical techniques in both the laboratory and the field will enhance our ability to connect the lab to the field.*

*Molecular-level chemical models increasingly provide valuable insights into complex multiphase processes. For example, important insights into the nature of the chemistry occurring on polar stratospheric cloud materials were obtained from molecular dynamics modeling (Bianco and Hynes, 2006), and more recent studies have addressed gas-surface interactions and the roles of solvent molecules in small molecular clusters (Gerber et al., 2015; Fang et al., 2019; Yang et al., 2019). Whereas past computational methods only included a few solvent molecules, current dynamics models using state-of-the-art force fields can realistically simulate partitioning, surface adsorption constants, diffusion constants, and vapor pressures, representing an important point of contact to the physical chemistry and chemical physics communities (Tobias et al., 2013). We also note that machine-learning learning techniques are very recently being applied to molecular dynamics simulations, for example, to describe the interactions of $N_2O_5$ with liquid water (Galib and Limmer, 2021) and the dissociation of strong acids at aqueous interfaces (de la Puente et al., 2022). For establishing fundamental parameters that are experimentally challenging to measure, such as the likelihood that a collision of a molecule with a particle leads to uptake by the condensed phase (i.e., a mass accommodation coefficient), theoretical methods may be preferable to experiment in some situations."*

Line 373/374: Modern multiphase chemistry modelling can predict the composition of that filter sample by travelling box approaches. What about that ? This part with observational capabilities could be improved considering multiphase modelling.

Response: This is a good suggestion, which is now added to *Section 6.8 on Grappling with Chemical Complexity*. New text for the importance of multiphase modeling (line 513):

*"Models working over a wide range of spatial and temporal scales can help address issues in chemical complexity. As mentioned in the previous section, molecular dynamics calculations are becoming increasingly sophisticated. So too are multiphase kinetics models that can incorporate insights gained at the molecular level into modelling frameworks that aim to couple the gas phase and condensed phases, including bulk reaction kinetics, mass transfer and interfacial processes (Poschl et al., 2007; Tilgner et al., 2013; Woo and McNeill, 2015). A challenge is to couple bulk and interfacial processes correctly. As computing capabilities grow, the complexity of the multiphase and detailed molecular mechanism models that can be incorporated into chemical transport models will also increase. Also, Lagrangian-type models increasingly can model specific field observations (Zaveri et al., 2010). We note that a successful hierarchical approach has arisen in the indoor chemistry community where modeling groups using a wide range of tools, from molecular dynamics to large-scale computational fluid dynamics, interact closely with each other and with experimental scientists (Shiraiwa et al., 2019)."*

Line 383 In this part, there are many specific examples but the modelling part is of a quite limited scope. It would be good to strengthen and also link to the preceeding observational capabilities section. Maybe these examples can be kept and discussed but where is a a section dealing with mechanism development, mechanism reduction, automated mechanism generation ? Assess the state of models. How to improve ? Do we have enough computer capacity - Moore's law still holds. And, then, for what ?

Response: This is a good point, partially addressed in the new text described in the previous response. However, just as we don't have the space to go into specific details about future instrumental approaches, we likewise do not do that for specific modeling methods.

Maybe a few more thoughts: There is again and again re-investigation of certain issues which are well implemented into leading models and everybody could look up what certain processes could do – look at S(VI) production in China – how many claims to explain did we see in the last decade ? I 'd say, 100. Fm NO2 to TMI, the uncatalyzed S(IV) oxidation, photosensitization, nitrate photolysis with artificial enhancement - you can find everything. But: problem solved ? No, they are all not doing the job.

Response: We agree.  Concerning the specific issue of S(VI) production in China, the following sentence was included in the original manuscript: *"An accurate quantitative assessment of these and other reaction pathways is still developing (Liu et al., 2021b)."* which we have now changed to (line 245):

*"An accurate quantitative assessment of these and other reaction pathways is still developing but far from complete (Liu et al., 2021b)."*

But there is exciting new fields (besides those you mention) : Continental halogen multiphase chemistry. Release of hydrated formaldehyde into the gas phase fm aqueous systems. Hydroperoxide and peroxyl radical chemistry. Role of TMI. Coupling of cloud process to aerosol composition (cf. sulfate as a role model !) Accretion product formation in gas and particle phase. Is there interesting NOx(aq) chemistry ? Condensed phase photochemistry. There is stuff where we must get better: DMS oxidation, aromatics oxidation, multiphase mechanism kernels beyond C4. Having an up-to-date versatile gas-phase backbone mechanism to which everything couples. Have multiphase chemistry mechanisms with interfacial and bulk parts coupled correctly.

Response: These are excellent suggestions. However, we note that in the final section of the manuscript (*Section 7. Concluding thoughts*) we intentionally did not mention specific chemical systems that require additional work because there are so many to mention!  That said, we have addressed some of the more general suggestions:

TMI See Response above concerning TMI.

Condensed phase photochemistry This topic is covered in *"Section 6.5 Understanding the role of light"*

Coupling of cloud processes to aerosol composition New text (line 369): *"In addition to continuing to address the fundamentals of cloud chemistry oxidation processes, the associated chemistry of transition metals, and the production of oxidants within cloudwater and via uptake from the gas phase (Herrmann et al., 2015), there is a particular need to also study such processes at cold temperatures, including under supercooled water conditions."*

and (line 222)

*"In addition to forming organic aerosol via gas-to-particle conversion, they are produced from the evaporation of cloud droplets. Oxidation processes occur within cloud droplets (Herrmann et al., 2015), producing more oxidized organics that yield oxygenated aerosol upon evaporation. Similar reactions, proceeding at much higher organic reactant concentrations, can also occur within the aqueous component of tropospheric aerosol (Blando and Turpin, 2000)."*

Coupling of interfacial and bulk chemistry  New text (line 514): *"So too are multiphase kinetics models that can incorporate insights gained at the molecular level into modelling frameworks that aim to couple*

*the gas phase and condensed phases, including bulk reaction kinetics, mass transfer and interfacial processes (Poschl et al., 2007; Tilgner et al., 2013; Woo and McNeill, 2015).*